# A Rare Case of Effusive-Constrictive Pericarditis Caused by *Streptococcus agalactiae*: Emergency Surgical Treatment

**DOI:** 10.3390/medicina58060699

**Published:** 2022-05-25

**Authors:** Annarita Iavazzo, Giovanni Battista Pinna, Maria Grazia Romeo, Emilio Mileo, Emanuele Pilato, Luigi Di Tommaso

**Affiliations:** Adult and Pediatric Cardiac Surgery, Department of Advanced Biomedical Sciences, University of Naples Federico II, 80131 Naples, Italy; giovannipinna@alice.it (G.B.P.); mariagrazia2791@gmail.com (M.G.R.); emiliomileo@gmail.com (E.M.); emapilato@yahoo.it (E.P.); lditommaso@tin.it (L.D.T.)

**Keywords:** pericarditis, *Streptococcus agalactiae*, pericardial effusion

## Abstract

A 70-year-old male patient was admitted to the emergency room in cardiac arrest. The patient was resuscitated and then referred to our cardiac surgery department, where he was diagnosed with suspected effusive constrictive pericarditis. A failed trial of TEE-guided pericardiocentesis led to the decision of surgical intervention. Sternotomy was performed and revealed pericardial thickening and very dense adhesions involving the pericardium and both pleurae, suggesting a neoplastic disease. An extensive pericardiectomy and bilateral pleural decortication were performed. After surgery, the patient improved significantly and was discharged from the intensive care unit 24 h later. Pericardial thickening, dense adhesions, the amount and color of pericardial fluid and the aspect of epicardial tissue increased our suspicion of neoplastic disease. Histological samples were sent to be analyzed immediately; a few days later, they were unexpectedly negative for any neoplastic disease but showed a group-B-hemolytic *Streptococcus agalactiae* infection, which causes pericarditis in extremely rare cases. Postoperatively, the patient, under intravenous antibiotic and anti-inflammatory therapy, remained asymptomatic and was discharged ten days after the surgery. At the three-month follow-up, transthoracic echocardiography showed a normal right and left ventricular function with no pericardial effusion.

## 1. Introduction

Effusive-constrictive pericarditis (ECP) [1] is a rare condition that was first described by Hancock in 1971 as a distinct clinical and hemodynamic condition in which constriction of the heart by the visceral pericardium occurs in the presence of tense pericardial effusion. Both of these conditions cause a tamponade-like effect on the heart.

Hemodynamical changes are caused by the pericardial thickness: the heart is constricted by a rigid fibrinous or calcified pericardium, and the ventricular filling is limited during mid or late diastole.

Most cases of ECP are idiopathic; other causes include infections (most commonly *Mycobacterium Tuberculosis*), radiation, malignancy, chemotherapy and post-surgical pericardial disease.

Its diagnosis is not always easy, and it requires a detailed analysis of a variety of investigations. Clinical symptoms can include distended jugular veins (because of high pressure), Beck’s triad (hypotension, pulsus paradoxus and muffled heart sounds), the Kussmaul sign, lethargy, ascites, and lower limb edema.

The initial diagnostic methods include electrocardiography, echocardiography and chest CT.

## 2. Case Report

A 70-year-old male with type 2 diabetes on Insulin therapy, chronic ischemic heart disease, and a 1-month history of right heart failure symptoms presented with an isolated episode of syncopal attack associated with hypotension.

Four weeks before hospital admission, the patient suffered from progressive dyspnea, orthopnea, and perimalleolar edema, despite being under diuretic treatment (furosemide 75 mg daily). A physical examination revealed pallor and dryness of the skin and the mucosae, distended jugular veins, signs of Kussmaul, perimalleolar edema, low-grade fever (37.3 °C), and cachexia. His blood pressure was 100/60 mmHg. An objective thoracic examination revealed muffled heart sounds, pulsus paradoxus, and pulmonary auscultation revealed absent vesicular breath sound on the left lung base and pulmonary crepitations on the right lung base. Electrocardiogram revealed sinus rhythm at a rate of 82 beats/min and diffuse ST elevation.

The blood tests showed an elevated white blood cell count (WBC 12,500/μL), anemia (11 g/dL), renal insufficiency (Crea 1.81 mg/dL; Urea 112 mg/dL; eGFR 39.5 mL/min), hepatic dysfunction (increased transaminases, AST 1183 U/L; ALT 960 U/L; increased LDH 1830 U/L), INR 2.70, C-reactive protein 9 mg/L; NT-proBNP 437 pg/mL.

The echocardiogram on admission showed massive circumferential fibrinous-organizing pericardial effusion (2.5 cm at the inferior wall of the right ventricle and 1.3 cm at the lateral wall of the left ventricle), severely compressing the right atrium (RA) but with no right ventricle (RV) collapse, impaired ejection fraction (EF 45%), dilated inferior vena cava (IVC 29 mm), increased pulmonary artery systolic pressure (PAPs 40 mmHg), and moderate left pleural effusion (Figure 1).

Total-body contrast-enhanced computed tomography (CT) was performed, which showed large pericardial effusion with hyperemia of the pericardial layers, associated with moderate bilateral pleural effusion, more significant on the left side (Figure 2).

Oncological markers (alpha fetoprotein, CA 19.9, and carcinoembryonic agent) were tested and were all negative.

On the second day of admission, a failed attempt of transesophageal echocardiography-guided pericardiocentesis with invasive cardiac monitoring was undertaken.

Intravenous diuretic treatment (furosemide 500 mg day) and anti-inflammatory therapy (prednisone 100 mg daily and indomethacin 100 mg daily) were started. Despite immediate and appropriate medical treatment, the clinical condition of the patient worsened. Another transthoracic echocardiogram showed a worsening of the pericardial effusion when the diameter (3.4 cm) increased. After another episode of severe hypotension, the patient was referred for immediate surgical treatment.

A pericardiotomy via median sternotomy approach was performed, revealing 800 mL of sero-hematic fluid. The epicardial surface showed a shaggy fibrinous exudate, a rare manifestation of fibrinous pericarditis (Figure 3). Decortication started from the left ventricle to the right ventricle to avoid pulmonary edema; after freeing the mid-anterior part, the dissection was extended laterally to both the right and left side using sharp and blunt techniques, carefully finding a dissection plane between the epicardium and the pericardium. The dissection was completed by resectioning a particularly tough adherence involving the cardiac apex. A phrenio-phrenic pericardiectomy was not considered necessary; furthermore, it could even be dangerous. The left pleura was opened, and the effusive pleural liquid was drained. Cultures of pericardial and pleural fluid were sent to be analyzed. The pericardial and pleural fluid was positive for *S. agalactiae*. The anatomo-pathologic exam confirmed the diagnosis of fibrinous pericarditis, showing fibrinous-granulomatous tissue associated with the presence of leukocyte exudate and blood clots.

Based on the antibiogram, the patient underwent an antibiotic treatment of benzylpenicillin with diuretic and anti-inflammatory therapy for two weeks.

The patient was asymptomatic and free from any surgical complications at the three-month follow-up after surgery.

## 3. Discussion

Effusive-constrictive pericarditis is a rare condition in which constriction of the heart by both the visceral pericardium and pericardial effusion occurs.

The diagnosis is not always straightforward, and sometimes the etiology cannot be identified.

In the modern antibiotic era, the *S. agalactiae* infection [2,3,4] of the endocardium and pericardial space is an extremely rare occurrence (Table 1). Predisposing factors for an *S. agalactiae* infection are related to immunosuppressive conditions, such as alcoholism, diabetes mellitus, liver cirrhosis, intravenous drug abuse, neoplasms, and HIV infections. The particular aspect of the fibrinous pericarditis with fibrin strands and acute inflammatory cells infiltrate is explained by the inability of *S. agalactiae* to produce fibrinolysis, leading to large vegetations, which gave this patient the macroscopic morphology of a shaggy heart [5,6].

Some blood clots were found in the pericardial fluid, and iatrogenic mild self-limiting bleeding that could have occurred during the failed pericardiocentesis cannot be excluded.

A traditional pericardiectomy was performed successfully. Unfortunately, some patients need a more aggressive surgical treatment, such as the Waffle technique [7], but it was not necessary in our case.

A rapid accurate diagnosis, surgical intervention and subsequent massive intravenous antibiotic and anti-inflammatory therapy led to complete healing of the patient in two weeks.

## Figures and Tables

**Figure 1 medicina-58-00699-f001:**
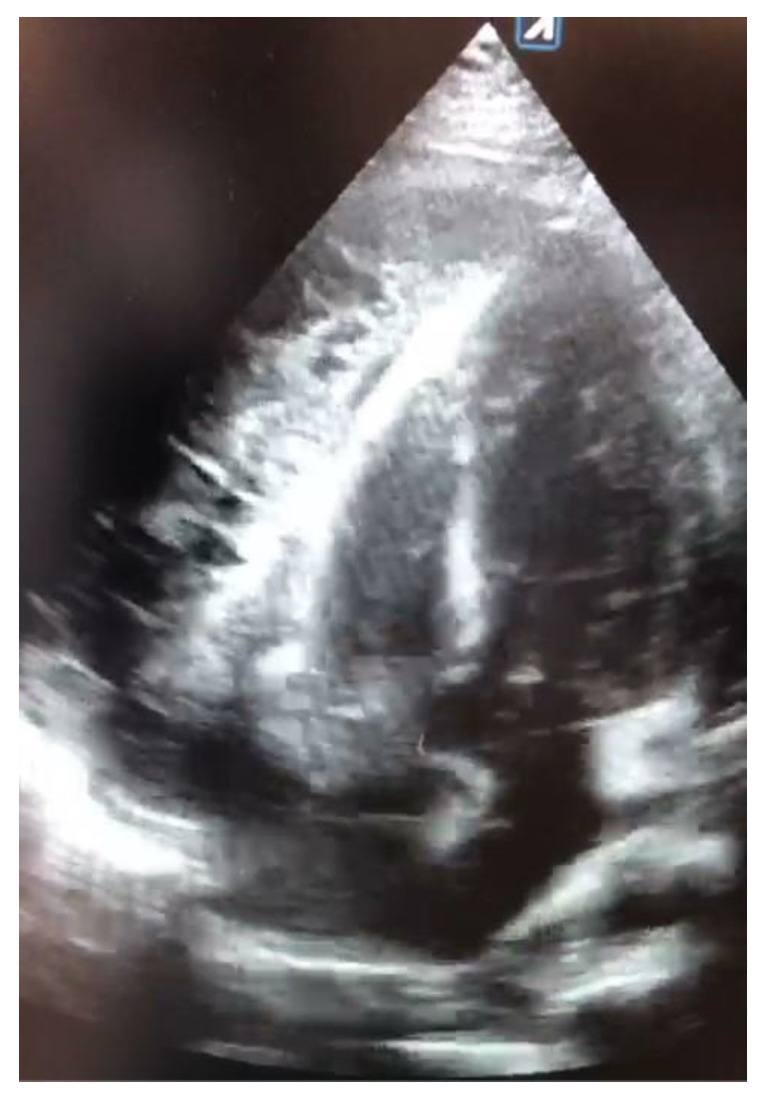
Transthoracic echocardiogram on admission showed large pericardial effusion with several fibrous bands.

**Figure 2 medicina-58-00699-f002:**
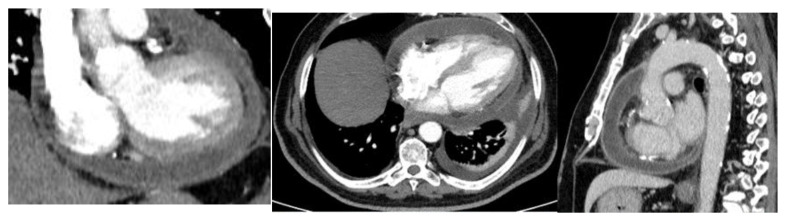
Contrast enhanced in thoracic CT. Severe pericardial effusion and left pleural effusion were observed.

**Figure 3 medicina-58-00699-f003:**
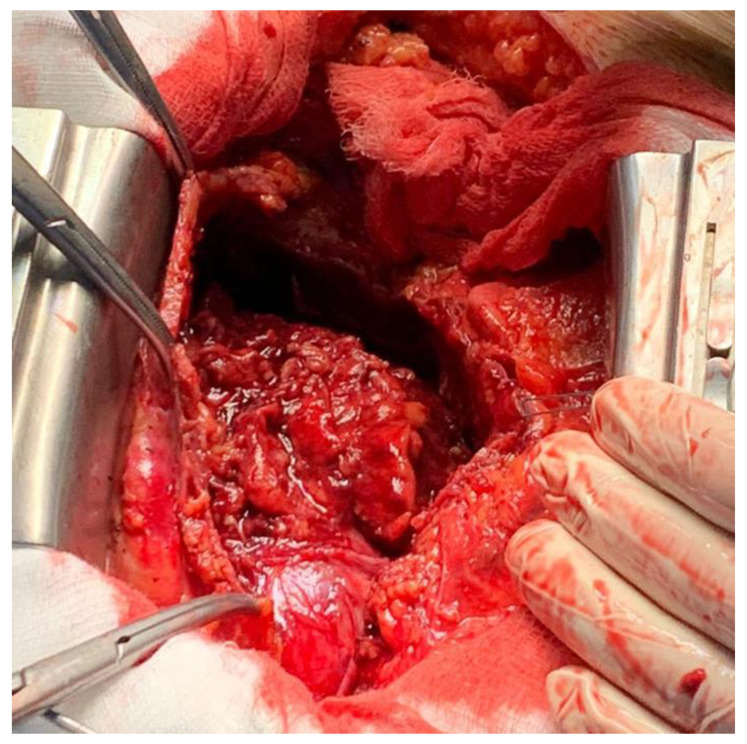
Effusive-constrictive pericarditis. Intraoperative image shows residual pericarditis after pericardiectomy.

**Table 1 medicina-58-00699-t001:** Reported cases of *S. agalactiae* pericarditis.

Age-Sex	Description	Clinical Features	Treatment	Result	References
61-F	Purulent pericarditis	DM II; right ventricular diastolic collapse	pericardiocentesis	Exitus	[7]
46-F	Purulent pericarditis	DM II, septicemia, cardiac tamponade	Pericardial window	Discharge	[6]
65-F	Purulent pericarditis-infective endocarditis	Septicemia, right ventricular diastolic collapse	Pericardiocentesis	Exitus	[4]
49-M	Effusive-constrictive pericarditis	DM II, septicemia, acute renal failure, cardiac tamponade	Pericardiocentesis + pericardiectomy (Waffle tecnique)	Discharge	[8]
46-F	Purulent pericarditis	Septicemia in sickle cell disease	Pericardial window	Discharge	[2]

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
