# Peer review of "A Rare Case of Effusive-Constrictive Pericarditis Caused by Streptococcus agalactiae: Emergency Surgical Treatment"

_medicina, 2022, doi:10.3390/medicina58060699_

Round 1
Reviewer 1 Report
You presented an interesting case report. However, certain modifications to the manuscript are necessary.
The patient's deterioration occurred after unsuccessful pericardiocentesis, and blood clots were described on pathological examination. You should clearly state that cardiac surgery was probably induced with iatrogenic worsening of the pericardium effusion.
I do not see the necessity of describing colonoscopy in this case report. From my point of view it is irrelevant.
Author Response
Dear Reviewer,
Thanks for your comments. I provided to add an explanation note about your interesting comment in the paragraph "Discussion" of the paper. Despite the worsening of the pericardium effusion, the other authors and I don't think that a iatrogenic lesion has occurred. Infact, we didn't find any source of active bleeding. Anyway, as I explain in my reviewed version, we can't exclude a self-limiting bleeding.
I deleted the part about the colonscopy, it's irrelevant, as you noticed.
Thank you for your review.

Reviewer 2 Report
Firstname Lastname and colleagues showed that effusive-constrictive pericarditis caused by Streptococcus Agalactiae and surgical treatment effectiveness. In particular, the pericardial infection of S.Agalactiae is very rare and subsequent drastic hemodynamic pathology is very impressive.
Overall, the findings are interesting, and the data appear to be of good quality, but the authors need to present their findings in more detail.
Major and minor comments are as follows:
Major comments
・Authors showed pericarditis and pre-tamponade findings in this case report, however, didn't describe hemodynamical evidence of constrictive pericarditis. How did authors evaluate these pathology?
・Authors showed physical examination finding of pulsus paradoxus. This finding is rare in constrictive pericarditis because epicardial adhesions are stronger than cardiac tamponade. Was there a sign of Kussmaul?
Author Response
Dear reviewer,
thank you very much for your comments.
As you asked, I added a little explanation about the hemodynamical evidences of constrictive pericarditis. Otherwise, unfortunately, we couldn't perform a catetherism, so we don't have pictures of the changing in the waves on venous pressure of the patient. It would have been very interesting.
Yes, there was the sign of Kussmaul, I added it in the paper, as you suggested.

Reviewer 3 Report
Thanks a lot for the chance to review this case study. I think the manuscript is well written and the clinical details are clear.
Only two minor comments:
Line 39: "Four weeks before", does this mean before hospital admission? Also Line 72: "The day after", does it mean second day of admission?
Line 99: "Basing on", it should be "based on".
Author Response
Dear Reviewer, thank you very much for your comments.
I provided to correct the errors in the Line 39, 72, 99.
I really apologize for my mistakes and I thank you to make me notice them.

Round 2
Reviewer 1 Report
The authors partially accepted the recommendations. Extensive editing of English is mandatory.
Author Response
Dear Reviewer,
thank you for comments. I provided to an extensive editing of English language and style, as you suggested. I asked for help to an English colleague to revise the manuscript.
Please see the attachment.
Kind regards

Reviewer 2 Report
Thank you for responding to my comments. The authors have substantially improved the manuscript following input from the previous reviews. I think the readership will appreciate it.
Author Response
Dear Reviewer,
thanks for your comment.
This manuscript is a resubmission of an earlier submission. The following is a list of the peer review reports and author responses from that submission.